# Surface Modification and Mechanical Properties Improvement of Bamboo Fibers Using Dielectric Barrier Discharge Plasma Treatment

**DOI:** 10.3390/polym15071711

**Published:** 2023-03-29

**Authors:** Choncharoen Sawangrat, Parichat Thipchai, Kannikar Kaewapai, Kittisak Jantanasakulwong, Jonghwan Suhr, Pitiwat Wattanachai, Pornchai Rachtanapun

**Affiliations:** 1Department of Industrial Engineering, Faculty of Engineering, Chiang Mai University, Chiang Mai 50200, Thailand; 2Doctor of Philosophy Program in Nanoscience and Nanotechnology (International Program/Interdisciplinary), Faculty of Science, Chiang Mai University, Chiang Mai 50200, Thailand; 3Science and Technology Park (STeP), Chiang Mai University, Chiang Mai 50100, Thailand; 4Division of Packaging Technology, School of Agro-Industry, Faculty of Agro-Industry, Chiang Mai University, Chiang Mai 50100, Thailand; 5Cluster of Agro Bio-Circular-Green Industry (Agro BCG), Chiang Mai University, Chiang Mai 50100, Thailand; 6School of Mechanical Engineering, Sungkyunkwan University, 2066 Seobu-ro, Jangan-gu, Suwon-si 16419, Gyeonggi-do, Republic of Korea; 7Department of Civil Engineering, Faculty of Engineering, Chiang Mai University, Chiang Mai 50200, Thailand

**Keywords:** bamboo fiber, composite, surface modification, plasma treatment, dielectric barrier discharge (DBD) plasma

## Abstract

The effect of argon (Ar) and oxygen (O_2_) gases as well as the treatment times on the properties of modified bamboo fibers using dielectric barrier discharge (DBD) plasma at generated power of 180 W were investigated. The plasma treatment of bamboo fibers with inert gases leads to the generation of ions and radicals on the fiber surface. Fourier transform-infrared spectroscopy (FTIR) confirmed that the functional groups of lignin and hemicellulose were reduced owing to the removal of the amorphous portion of the fibers by plasma etching. X-ray diffraction analysis (XRD) results in an increased crystallinity percentage. X-ray photoelectron spectroscopy (XPS) results showed the oxygen/carbon (O/C) atomic concentration ratio increased with increasing treatment time. The fiber weight loss percentage increased with increased treatment time. Scanning electron microscopy (SEM) images showed that partial etching of the fiber surface led to a higher surface roughness and area and that the Ar + O_2_ gas plasma treatment provided more surface etching than the Ar gas treatment because of the oxidation reaction of the O_2_ plasma. The mechanical properties of fiber-reinforced epoxy (FRE) matrix composites revealed that the F_(tr)_RE-Ar (30) samples showed a high tensile strength, whereas the mechanical properties of the F_(tr)_RE-Ar + O_2_ sample decreased with increased treatment time.

## 1. Introduction

Plasma is normally produced by the distinct separation of highly charged positive ions and negatively charged electrons to create an electric field that produces electric currents and magnetic fields [1]. Plasma contains highly charged particles such as electrons, ions, radicals, and neutral atoms that can be excited and ionized [2]. These reactive species can break the reactant molecules or change the surface structure of the material, depending on the composition of the material and the plasma conditions [3].

Plasma technology has become an active, high-growth research field in recent years, and it is widely used in the textile industry (natural protein, cellulose, and synthetic fibers) among all available material surface modifications [4,5]. Plasma technology can produce a series of cleaning, etching, polymerization, cross-linking effects, as well as other complex physical and chemical effects [6,7]. Normally, the interaction of plasma affects only a very thin surface layer, where photons can modify the surface to a depth of a few nanometers to several hundred nanometers [1,8]. The advantages of this method are that it is a dry treatment and is considered a promising and environmentally friendly method because it does not use harmful liquids or gases and leaves little or no waste [9,10]. Four types of cold plasma discharge methods are commonly used for fiber-reinforced composites: corona discharge, atmospheric-pressure glow discharge (APGD), dielectric barrier discharge (DBD), and atmospheric-pressure plasma jet (APPJ) [11,12].

Bamboo is the plant that grows the quickest on Earth, reaching full maturity within three to five years [13,14]. Bamboo forests are abundant in Asian and South American countries [15]. Bamboo is widely used for furniture, handicrafts, flooring, interior decoration materials, composite materials, and civil building materials [13,16]. Therefore, a significant amount of bamboo fiber waste is generated during the machining process. An alternative method for bamboo fiber waste disposal is to use bamboo fibers as a polymer mixture to reinforce composite materials, which would alleviate the environmental burden of bamboo waste and provide economic benefits [13,17]. Bamboo fibers (BFs), have the chemical composition of BFs as cellulose (36.8–54.9%), hemicellulose (62.0–79.9%), lignin (23.1–33.5%), and waxes (1.93–7.95%) [18,19]. The extractive-free bamboo can be converted to micro- or nanoscale cellulose materials by a chemical treatment process. The 2,2,6,6-tetrametylpiperidine-1-oxyl (TEMPO)-mediated oxidation system is one potential and efficient process to extract holocellulose from the original plant [20]. The BFs were oxidized using a TEMPO oxidation; when oxidation occurs, they mostly disintegrate into individual holocellulose after being removed from the undesired fraction [21]. The hydroxyl groups of bamboo cellulose can be bonded with a functional monomer or copolymer, and bamboo cellulose itself has a high crystalline cellulose volume, which provides the fibers with very high tensile strength and modulus [22]. Therefore, they are used in a variety of applications, particularly in the textile industry. Recently, natural fibers have received much attention as reinforcements for composite materials to replace glass and carbon fibers because of their low cost, lightweight, high strength and stiffness, renewability and biodegradability, and low impact on the environment because they reduce the use of fossil energy and the release of carbon dioxide into the atmosphere [13,15,23].

However, the interfacial adhesion between BFs and the composite matrices is poor because of the mismatch between the high hydrophilicity of BFs and the hydrophobicity of the polymer matrices, resulting in poor performance of the composites [24]. The compatibility, interfacial adhesion, and reinforcement capability of natural fibers and the matrix are related to the surface energy and specific surface area of the fibers, and interfacial bonding between the fiber and matrix impacts the performance of the composites [25]. Therefore, surface improvement of the fibers is important for increasing the compatibility between the fibers and the polymer matrix to improve the mechanical properties of the composites. Many techniques have been applied to modify fiber surfaces to improve the adhesion properties between the fibers and the matrix. Common chemical methods include alkaline treatment, acetylation, benzoylation, treatment with silanes, and the addition of coupling agents, among others [26]. Physical methods include plasma treatment, γ-ray irradiation, corona treatment, and ultraviolet (UV) irradiation [27,28].

DBD plasma is one of the most effective nonthermal atmospheric plasmas; thus, it is preferred for low-voltage applications [29]. It is widely used for surface modification in the textile and polymer composite industries [30,31]. DBD is a class of plasma source that provides an insulating cover over the electrodes [32]. The specific advantages of atmospheric-pressure processing techniques, such as low-pressure treatment, offer much better stability, control, reproducibility, easy formation of a stable discharge and discharge scalability, and elimination of expensive vacuum equipment [30,33,34]. According to recent research, atmospheric plasma treatment is useful for modifying surface properties such as wettability, surface energy, and surface morphology without affecting the bulk properties of the material [35,36]. This surface modification method can remove lignin, hemicellulose, waxes, and other extractable substances, which increases the fiber surface roughness and contact area, thus improving the compatibility and interfacial adhesion between the natural fibers and the polymer matrix [37,38,39].

This study focuses on the preparation and modification of BFs using DBD plasma treatment, and the various properties of the BFs before and after plasma treatment were investigated. The morphology and microstructure of the fiber surfaces were examined using scanning electron microscopy (SEM). Fourier transform-infrared (FTIR) spectroscopy was used to characterize the surface chemical properties of the BFs [40]. The chemical bonds were analyzed using X-ray photoelectron spectroscopy (XPS). X-ray diffraction (XRD) analysis was used to analyze the crystallinity [41]. The specific surface areas were determined using the Brunauer–Emmett–Teller (BET) method and the mechanical properties of the bamboo–FRE matrix composites were evaluated [42]. In addition, a model of the relationship between the response variables and experimental factors was determined using response surface analysis.

## 2. Materials and Methods

### 2.1. Materials

Bamboo trunks (Bambusa longispatha) were obtained from the Samoeng District, Chiang Mai Province, Thailand. Acetic acid, sodium hydroxide (NaOH), and sodium chlorite (NaClO_2_) were purchased from Merck & Co., Inc. (Darmstadt, Germany) and used for the alkaline treatment and sodium chlorite bleaching processes, respectively. All chemicals were of reagent grade and used as received.

### 2.2. Materials Preparation

The bamboo fibers were peeled off to obtain a pure trunk, which was cut into pieces (2 × 6 in). These pieces were then dried in a hot air oven at 80 ± 3 °C for 12 h. Later, they were ground into a rough powder by using a grinder (Grinder ML-SC5-III, Ming Lee Industrial Ltd., Hong Kong, China). The ground bamboo was dried in a hot air oven at 105 ± 3 °C for 6 h. Next, the bamboo particles were crushed with a high-speed blender (Dxfill machine, DXM-700-F, Shanghai, China) at a speed of 35,000 rpm for 15 min. The bamboo powder was dried in a hot air oven at 80 ± 3 °C for 12 h. The powder was weighed until it was stable by measuring the moisture of dried bamboo powder to less than 1%. The percent dryness (%) of the dry bamboo powder was calculated using Equation (1).
(1)$$\%\;Dryness=\frac{Weight\;of\;cellulose\;without\;moisture\;content}{Weight\;of\;cellulose\;content}\times 100$$

### 2.3. Preparation of Cellulose from Bamboo

In the alkaline treatment process, the bamboo powder (100 g) was placed in a beaker and treated at 80 °C with 20% *w*/*v* NaOH (2000 mL) under continuous stirring (Bethai Bangkok Equipment & Chemical Co., Ltd., Bangkok, Thailand) at 1000 rpm for 5 h. The extracted bamboo was filtered and washed with distilled water until the pH became 7. The product was then dried in an oven at 80 °C for 12 h. The bamboo pulp was bleached with NaClO_2_ to remove hemicellulose, lignin, and other noncellulosic substances from the alpha-cellulose. The dried pulp (100 g) was mixed with an acetate buffer comprising 5.4% NaOH *w*/*v* (850 mL) and 150 mL of acetic acid in 1000 mL of distilled water. The mixture was boiled in 3.4% NaClO_2_ (1000 mL) at 85 ± 5 °C for 3 h with continuous stirring at 500 rpm [43]. The final product was then filtered and washed with distilled water until the pH became 7. The cellulose obtained from the bleaching process was dried in a dry oven at 80 ± 3 °C for 12 h. The bleaching procedure was repeated twice. Finally, the cellulose was stored in a desiccator.

### 2.4. Chemical Constituents Analysis

The amount of holocellulose was obtained from the extractive-free bamboo using the T203 om-99 acid chlorite test method. The obtained holocellulose was extracted using 17.5% *w*/*v* sodium hydroxide, following TAPPI T203 om-93, to remove hemicellulose and leave only alpha-cellulose. The extractives, which were wall substances in cells mainly consisting of ground bamboo, were analyzed according to TAPPI T204 om-97, 95% ethanol extraction (TAPPI T264 om-97), and hot water (TAPPI T207 om-93) before composition analysis. Extractive-free fibers were extracted using a 24 N solution of H_2_SO_4_ following TAPPI T222 om-98 to determine the lignin content of the residue that was not hydrolyzed by acid. These measurements were repeated three times for each sample and the average value is reported.

### 2.5. Plasma Treatment Process

A three-level full factorial design was used for the plasma treatment of the extracted BFs. The samples were randomly divided into two groups: an untreated group and a group treated with different plasma gases for various treatment times. A schematic of a fiber treated with the DBD plasma system is shown in Figure 1. The first step was to place the extracted bamboo fiber on an aluminum tray and then treat it with DBD plasma (Mini-smart, Republic of Korea, www.plasmart.com, accessed on March 2022). DBD plasma was generated between two parallel electrodes: a powered electrode and a grounded electrode. A schematic of the atmospheric-pressure DBD cell is shown in Figure 1. The electrodes were connected to an RF power supply source at a frequency of 13.56 MHz to ensure plasma radiation exposure, the sample tray was moved back and forth under the grounded electrode at a rate of 30 cm/s. The distance between the powered electrode and the sample tray (discharge gap) was set to 1 mm. In this study, the plasma discharge power was set to 180 W (3.45 W/cm^2^). When a plasma system is operated at a low plasma frequency, longer wavelengths are created that give ions a large amount of kinetic energy. This higher ion density results in the sample position in this space everywhere will have a similar result. Ar and O_2_ gases were used, with an Ar flow rate of 8 L/min and an O_2_ flow rate of 10 L/min. The treatment times were 10, 20, and 30 min. After completing the treatment under each condition three times, the samples were stored in a foil bag before analyzing their various properties. The plasma-treated samples were labeled after each plasma treatment as shown in Table 1.

### 2.6. Fourier Transform-Infrared Spectroscopy (FTIR)

The functional groups of the untreated and treated plasma bamboo fiber samples were analyzed using an FTIR spectrometer (FT/IR-4700, JASCO International Co., Ltd., Pfungstadt, Germany), Spectra Manager™II software for data processing and instrument control. The samples and KBr were mixed and prepared as sheets by compression in a sample holder. The FTIR spectra were measured over a range of 500–4000 cm^−1^.

### 2.7. X-ray Photoelectron Spectroscopy (XPS)

The surface chemical composition was analyzed using XPS (AXIS Ultra ^DLD^, Kratos Analytical Co., Ltd., Manchester, UK). The XPS parameters were a monochromatic Al X-ray at 150 W anode power and survey spectra from 0 to 1200 eV with a pass energy of 160 eV for a full survey and 40 eV for the core-level spectra. The plasma-treated bamboo fiber powder was dispersed onto carbon tape, which was then placed on a stainless-steel sheet in the ultrahigh vacuum chamber for XPS analysis. Data acquisition and processing were performed using ESCApe software VISION II by Kratos analytical Co., Ltd. 

### 2.8. X-ray Diffraction Spectroscopy (XRD)

The crystallinity of the BFs was analyzed using XRD (SmartLab X-ray Diffractometer (Rigaku, Ltd., Tokyo, Japan), MDI JADE (6.8.0) software was used for data analysis. The XRD patterns were measured in the range 2θ = 10° to 60°. The fiber crystallinity index (C.I.) was calculated using Equation (2).
(2)$$\%\;C.I.=\frac{I_{002}-I_{\mathrm{am}}}{I_{002}}\times 100$$
where I_002_ is the maximum intensity of the 002 crystalline peak and I_am_ is the minimum intensity of the amorphous material between the 101 and 002 peaks [44].

### 2.9. Scanning Electron Microscopy (SEM)

The microstructures of the plasma-treated bamboo fiber samples were examined using SEM (JSM-IT300, JEOL., Ltd., Tokyo, Japan). The fiber samples were placed on metal stubs with carbon tape and then sputter-coated with an Au film for 45 min prior to SEM analysis. The acceleration voltage was 10 kV at 1000× and 15,000×. Microstructural characterization was used to reveal the differences between the physical properties of the bamboo fiber samples before and after plasma treatment.

### 2.10. Brunauer–Emmett–Teller (BET)

The fiber surface areas before and after plasma treatment were determined using the BET equations, which were the values obtained from a Quantachrome apparatus (Nova Instruments, Anton Paar Quanta Tec Inc., Boynton Beach, FL, USA). Data acquisition and analysis were performed using Quantachrome NovaWin software version 11.06. Three samples were analyzed from each condition.

### 2.11. Mechanical Properties

Bamboo-fiber-reinforced epoxy matrix composite (FRE) samples were prepared for tensile testing using a casting process. The composite samples were prepared using untreated and plasma-treated BFs (F_tr_) mixed with an epoxy matrix. The amount of fiber used to prepare the composite mixtures was 5 wt% of the sample. The labels and compositions of the FRE samples are listed in Table 2. The composite fluid was then poured into a mold using the vacuum technique. The dimensions of the bone-shaped samples were measured according to the JISK-6251-7 standard. The tensile tests were performed using a universal testing machine (MCT-1150, Hounfield Test Equipment, Surrey, UK) at a crosshead speed of 10 mm/min as per the JIS standard, MCT-Logger 7Pro (32bit version) was used for data analysis. The tests were repeated five times for each sample to determine the tensile strength, elongation at break, and energy absorption.

### 2.12. Statistical Analysis

Data were analyzed using one-way analysis of variance (ANOVA), followed by Duncan’s multiple range test. The different letters are the significant level of *p* ≤ 0.05 by LSD test. Statistical analyses were performed using IBM SPSS software version 26.

## 3. Results and Discussion

### 3.1. Chemical Constituents of Extracted BFs

Table 3 presents the chemical constituents of bamboo. The BFs consisted of 41.67% alpha-cellulose, 73.10% holocellulose, 28.88% lignin, 3.17% extractive, and 2.04% ash, as previously reported [19]. The chemical constituents of the bamboo are slightly different from those previously reported because of the difference in species, but there is generally good agreement [45]. This can be explained by analyzing the morphology of the BFs and parenchyma cells, which show the morphology of several macerated BFs [46].

### 3.2. Morphology of Extracted BFs

Figure 2 shows the macroscopic images of the bamboo at different stages: after the crushing process (Figure 2a), the pulp after the alkaline treatment process (Figure 2b), and the cellulose after the bleaching process (Figure 2c). Figure 2c clearly shows the micromorphology of the cellulose. The fiber particles consist of both irregularly shaped and long, stick-like fragments. The average size of the bamboo particles was 90 μm whereas that of the pulp after the alkaline treatment process was 60–50 μm, and that of the cellulose after bleaching was 10–50.4 μm. It was also observed that the cellulose fibers became thinner and cleaner after treatment with sodium hydroxide and chlorite solutions (Figure 2b,c), respectively.

### 3.3. Analysis of Reactive Radical Species via Optical Emission Spectroscopy

The plasma species generated in the DBD plasma were determined using optical emission spectroscopy (OES) under different plasma gas operational conditions. The OES spectra for the argon gas plasma and Ar + O_2_ gas plasma in the wavelength range of 200–900 nm are shown in Figure 3a,b, respectively. The main peak emitted by the excited atoms of the feed gas (argon) was between 300 and 900 nm. Peaks corresponding to N_2_ were detected between 330 and 400 nm and were presumably presented because of the mixing of argon gas with nitrogen in the atmosphere surrounding the DBD plasma. Oxygen atomic peaks are observed at 406.3 and 777.1 nm for the argon gas plasma, but those peaks are higher for the Ar + O_2_ plasma treatment (Figure 3b). In addition, hydroxyl (OH) radicals are observed at 309.7 nm, probably as a result of O_2_ and H_2_O isolation [47,48,49]. Inert gas plasma treatment initiates surface activation by generating reactive species (ions, radicals, etc.,) on the cellulose surface. Free radicals formed during the argon plasma treatment on the surface of the fiber reacted with atmospheric oxygen, nitrogen, and moisture [50]. Reactive radical species, reactive oxygen species (ROS), and reactive nitrogen species (RNS) interacted with the surface by cleaning, etching, and breaking bonds, and the recombination of these radicals allowed crosslinking at the treated surface of the materials [51].

### 3.4. Analysis of Functional Groups via FTIR

The FTIR spectra of the untreated and plasma-treated fibers are shown in Figure 4. The band in the region of 3200–3600 cm^−1^ is strong in cellulose and hemicellulose and corresponds to the hydrophilic hydroxyl (OH) stretching vibration [52,53]. The band at 2904 cm^−1^ is the C–H stretching vibration of the hydrocarbon structure in hemicellulose [54]. The band at 1640 cm^−1^ represents the adsorbed water molecules in noncrystalline cellulose. It is attributed to the H–O–H stretching vibration of the adsorbed water owing to the strong interaction between adsorbed water and the hydrophilic surface O–H group of cellulose because of the hydroxyl groups in cellulose [55,56]. The spectra at 1428, 1370, 1167, and 1056 cm^−1^ represent the C–H deformation (methoxyl group in lignin), C–H deformation (symmetric), C–O stretching of ester groups, and C–OH stretching vibration in cellulose, respectively [28,57,58,59]. The band at 895 cm^−1^ represents the C–H deformation in cellulose [60].

From the results, after plasma treatment, the BFs have functional groups and chemical bonds such as those of the untreated BFs. However, as the plasma treatment time increased to 30 min, the intensities of the FTIR spectra of the functional groups of hemicellulose, lignin, and noncrystalline cellulose decreased to 3420, 2904, 1640, 1428, and 1370 cm^−1^. Free radicals in plasma, especially oxygen radicals, are highly reactive, which is important for modifying cyclic cellulose chains. Because of the oxidation reaction by oxygen radicals, activated oxygen radicals may also interact with the hydroxyl groups of the lignocellulosic chain [61]. Surface etching by plasma resulted in the removal of hemicellulose, lignin, and noncrystalline cellulose, which is consistent with the SEM morphology and microstructure analysis results.

### 3.5. Determination of Element Composition and Chemical Bonds

A low-energy-resolution scan of the bamboo fiber revealed a surface composed of carbon and oxygen atoms (Table 4). This indicates that the fiber surface was predominantly composed of hydrocarbon compounds. DBD plasma treatment had a positive effect on the fiber surfaces compared with the untreated plasma fibers. The fibers treated with Ar + O_2_ gas plasma showed a higher O/C ratio than those treated with Ar gas plasma, confirming that the oxygen gas plasma increased the oxygen atomic content. As the treatment time increased, the O/C ratio also increased. The results showed an increase in oxygen content and a decrease in carbon content because plasma treatment with inert gas improved the surface activity of the fibers, allowing free radicals to react with oxygen to form peroxides, and the atomic percentage of carbon decreased as the atomic percentage of oxygen increased [11]. The presence of ROS caused an oxidation reaction that etched and oxidized the cellulose surface, resulting in the removal of hemicellulose, lignin, and amorphous portions of the fiber surface [8,37]. These results are consistent with the FTIR analysis results.

XPS analysis was performed to determine the elemental bonding of the carbon and oxygen atoms and to compare the energy levels and fiber intensities of the untreated and plasma-treated fiber samples (Figure 5). The XPS spectrum of carbon atoms on the fiber surface showed the highest intensity, where energy levels of approximately 285.0, 286.7, and 287.5 eV indicate the characteristics of C1s forming C–C or C–H, C–O, and C–OH bonds, respectively [8,62,63]. The C–OH bond is absent in the untreated plasma fibers, but it appears after the plasma treatment of the BFs. It was found that with every increase in treatment time, the intensity of the C–OH bonds increased, whereas the intensity of the C–C or C–H bond decreased because of the plasma etching of the hydrocarbon bonds in the fibers. Ar^+^ ions, which are highly energetic ionic species with sufficient energy to break the C–C or C–H bonds, subsequently react to form various oxygen functional groups on the treated fiber surface [61,64].

### 3.6. The Crystallinity Property

The results of the crystallinity analysis and the XRD patterns of the fibers, shown in Figure 6, show the typical crystal lattice of cellulose. The presence of two peaks of cellulose fiber is seen at 2θ = ~15.5° and ~22.8°, corresponding to (101) and (002) planes, respectively [44]. The (002) peak is the major crystalline peak of cellulose I. The low intensity of the peaks could be associated with the noncrystalline portion of cellulose and amorphous compounds in the fiber (lignin, hemicellulose, and wax) [37]. The presence of another typical peak of cellulose at 2θ = ~34.5° in the plane (040) is also observed. Analysis of the constant location of the peak indicated that the treatments did not modify the crystalline structure of cellulose. The crystallinity results indicated that the crystallinity of the plasma-treated fiber increased with increasing the treatment time as compared to the untreated fiber, which had a C.I. percentage of 59.65%. The treatment times of 10, 20, and 30 min for the Ar gas plasma-treated fibers revealed C.I. percentages of 59.79, 60.94, and 61.36%, respectively, whereas those of the Ar + O_2_ gas plasma-treated fibers had C.I. percentages of 61.63, 62.25, and 62.91%, respectively. The crystallinity of the fibers treated with Ar + O_2_ gas plasma was therefore higher than that of the fibers treated with Ar gas plasma. The increased C.I. can be attributed to the removal of the amorphous portions and noncellulosic components of the fibers, which led to an increase in the percentage of cellulose during the fiber surface modification process. These results are consistent with the SEM and FTIR results. Plasma-treated fibers with relatively high crystallinity are beneficial for the manufacture of biocomposites and can improve their mechanical strength [65].

### 3.7. Comprehensive Weight Loss Percentage

The weight of the treated fiber was significantly lower than that of the untreated fiber. Figure 7 shows the percentage of fiber weight loss versus treatment time for fibers treated with the two types of gas plasma. In the case of Ar gas plasma, the weight loss did not significantly increase with treatment time. The highest weight loss shown is 6.7% after 30 min, followed by 5.2 and 3.3% after 20 and 10 min, respectively. In the case of Ar + O_2_ gas plasma, the weight loss did not significantly increase with treatment time for 10 and 20 min, but it significantly increased with the 30 min treatment time. The highest weight loss was observed to be 9.8% after 30 min, followed by 6.6 and 4.2% after 20 and 10 min, respectively. This result can be attributed to the Ar plasma cleaning and slow etching of the fiber surface, whereas the Ar + O_2_ plasma etching was faster owing to the oxidation reaction of oxygen radicals, which resulted in oxidation of the fiber surface, thinning the fiber surface [1,66]. This result is consistent with the SEM results.

### 3.8. Morphology of Treated BFs

The morphologies of the untreated bamboo cellulose fibers and the fibers treated with different gas plasmas for various treatment times are shown in Figure 8 and Figure 9, respectively. The surface characteristics of the plasma-treated BFs are visibly different from those of the untreated BFs. The BFs treated with Ar and Ar + O_2_ gas plasma show partial etching of the fiber surface, which increased with increasing treatment time. The plasma treatment caused the etching of lignin and hemicellulose on the fiber surface, which was left over from the bleaching process, exposing the crystalline cellulose fibers. The plasma treatment with Ar + O_2_ gas resulted in more surface etching than that with Ar gas because of the oxidation–reduction of oxygen gas, which oxidized the fiber surface [67]. Furthermore, the fiber was composed of cellulose microfibrils linked with hemicellulose and lignin. When the lignin and hemicellulose were removed, microfibrils appeared in the cellulose, as shown in Figure 9, resulting in a higher surface roughness. The roughness and increased contact area improved the adhesion between the fiber and matrix interface, allowing the matrix to spread, penetrate, and interlock onto the fiber surface [28,68].

### 3.9. BET Specific Surface Area

The physical adsorption and desorption of nitrogen gas on the fiber surface were analyzed using the specific surface area as determined from the BET equations. The surface areas of the BFs treated with the different gas plasmas for various treatment times were higher than those of the untreated bamboo fiber samples (Table 5). The surface area after treatment with the Ar + O_2_ gas plasma showed a significant increase with increasing plasma treatment time (*p* ≤ 0.05) compared to that of the untreated fiber. The surface area of the BFs treated with the Ar + O_2_ gas plasma for 30 min had the largest surface area, which was statistically significant. This result is consistent with the SEM results. Etching of the fiber surface reduced the amount of lignin and hemicellulose, which increased the number of microfibrils, thereby increasing the roughness of the cellulose surface [68]. The higher the fiber surface area, the greater the fiber interface area, which resulted in better adhesion between the fiber and the matrix [69,70].

### 3.10. Mechanical Properties

Figure 10a shows the tensile properties observed using the tensile tester, where the FRE matrix composite fabricated using the untreated fibers was used as a control. The control showed tensile strength and elongation at a break of 36.94 MPa and 6.02%, respectively. The tensile strength, elongation at break, and energy absorption (EA) of the F_(tr)_RE composites fabricated using treated fibers were calculated and are reported in Table 6. The tensile strength of the plasma-treated F_(tr)_RE composites was significantly higher than that of the untreated composite (control). The tensile strength of F_(tr)_RE composites treated with Ar gas plasma increased with increasing treatment time; the F_(tr)_RE-Ar (30) composite exhibited a high tensile strength of 46.30 MPa. The Ar + O_2_ gas plasma F_(tr)_RE composites also showed a high tensile strength, particularly the F_(tr)_RE-Ar + O_2_ (10) composite, which decreased with increasing treatment time. Furthermore, the elongation at break of the F_(tr)_RE composites decreased as the treatment time increased, but it was not significantly different from that of the other treatments. The prolonged treatment time of the Ar + O_2_ plasma treatment affected the fiber structure, causing damage and cracking of the fiber. The oxidation reaction of oxygen gas resulted in the oxidation of the fiber surfaced, making it thinner, which led to the brittleness of the fiber [71]. The EA of the F_(tr)_RE composites during tensile strain is a highly desirable property, in addition to tensile strength. The EA was calculated from the area under the stress–strain curve, and the results are shown in Figure 10b. The EA of the Ar gas plasma F_(tr)_RE composites increased with increasing treatment time, in which the F_(tr)_RE-Ar (30) composite showed the highest EA of 129.22 MPa·mm/mm, which is a significant increase when compared with the FRE-untreated composite (98.25 MPa·mm/mm). Plasma etching of the fiber surface increased the roughness, resulting in better diffusion and penetration at the fiber–matrix interface and creating interlocking bonding [28,72]. Furthermore, Ar gas plasma cleaning of the fiber surface resulted in a decreased contact angle, which enhanced the polarity [73]. This improved the surface energy of the fiber surface, leading to good interface adhesion, allowing it to absorb energy and resulting in the gradual growth of cracks until final failure occurred by exceeding the fracture toughness [74,75,76]. Composites with a large area under the stress–strain curve are more effective energy absorbers. Therefore, the plasma-treated fibers improved the mechanical properties of the composite matrix. Although the EA of the Ar + O_2_ gas plasma F_(tr)_RE composites did not significantly decrease with increasing treatment time because of the brittleness of the fibers, the plasma etched the fiber surface, which thinned the fibers and led to the fracture.

Figure 11 shows the fracture characteristics of the untreated fibers and the fibers plasma treated with Ar gas and Ar + O_2_ gas plasma for various treatment times after use as reinforcements in the epoxy composites. The plasma-treated F_(tr)_RE composites show better interfacial bonding between the treated fiber and the epoxy matrix than the FRE-untreated composite. The plasma discharge enhanced the migration of free radicals, which increased the cleaning and etching of the fiber surface, decreased the extent of voids in the composites, generated links between the two different material phases, and increased their interfacial adhesion properties [11]. The fiber surface etching with plasma treatment exhibited increased surface roughness, which improved the interfacial adhesion between the fiber and matrix because of the interdiffusion of molecules between the matrix and the fiber, resulting in an increase in the EA of the F_(tr)_RE-Ar samples. However, the interfacial bonding of the F_(tr)_RE-Ar + O_2_ composites decreased with increasing treatment time, and the oxygen plasma-treated fibers showed brittle breakage because prolonged etching of the fiber surface by plasma discharge caused thinning of fibers, and thus they fractured.

### 3.11. The Effect of Plasma Treatment on the Fiber Surface on the Energy Absorption Property of the FRE Composite Samples

Figure 12a shows the two main factors influencing the response variable, EA. The effects of the gas type and treatment time on the EA values were not significantly different, but they did interact to affect the EA, as shown in Figure 12b. The EA value depends on the relationship between the gas type and treatment time. With increased treatment time, the mean EA of the Ar gas-plasma-treated F_(tr)_RE composites increased, whereas that of the Ar + O_2_ gas plasma-treated F_(tr)_RE composites decreased. The oxidation reaction of ROS resulted in faster etching and thinning of the fiber surface, which led to the brittleness of the fiber and the rapid propagation of cracks until final failure occurred.

## 4. Conclusions

Plasma technology was used to successfully modify the surface of bamboo fiber. The DBD plasma treatment improves fiber surface energy, interface adhesion, and compatibility, as measured by contract surface area, surface chemical composition, and tensile strength. The bamboo fiber was etched onto the fiber surfaces after the plasma treatment, which increased roughness and surface area in the BET result. Plasma treatment removes an amorphous portion of the fibers and results in an increase in their percentage of crystalline cellulose. Treating the fiber by plasma with Ar for 30 min was found to be the optimum condition because it improved the mechanical properties of the FRE composites. Plasma treatment can therefore facilitate high surface interactions and improve the interfacial adhesion between natural fibers and the particles of a polymer matrix. The surface treatments transform the fibers into raw materials for the manufacture of multi-benefit compounds and matrices commonly used in the biocomposites sector.

## Figures and Tables

**Figure 1 polymers-15-01711-f001:**
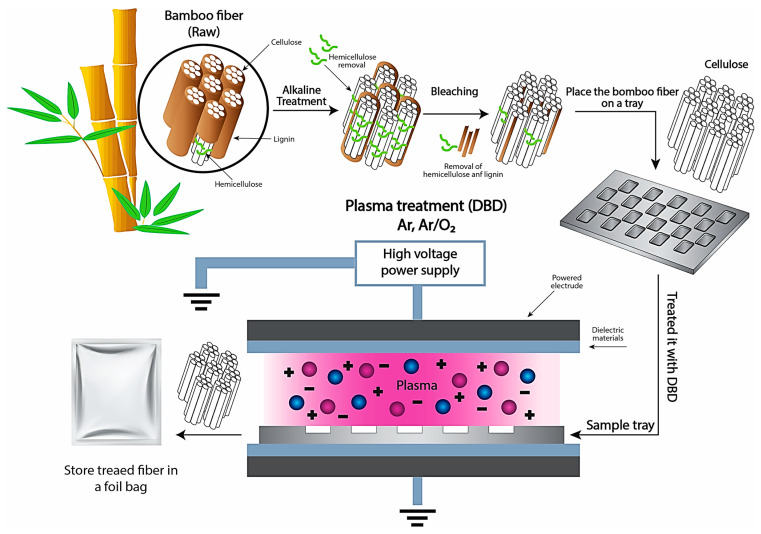
Schematic representation of the plasma treatment and diagram of atmospheric-pressure DBD plasma cell.

**Figure 2 polymers-15-01711-f002:**
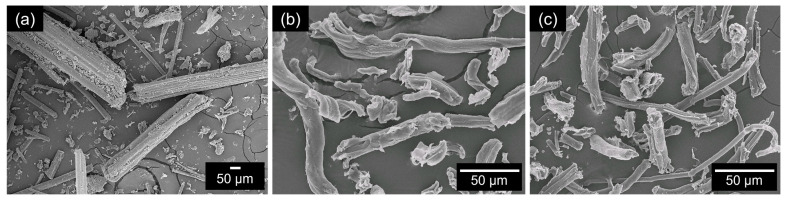
SEM images of (**a**) crude bamboo fiber at 100×, (**b**) pulp at 500×, and (**c**) cellulose at 500×.

**Figure 3 polymers-15-01711-f003:**
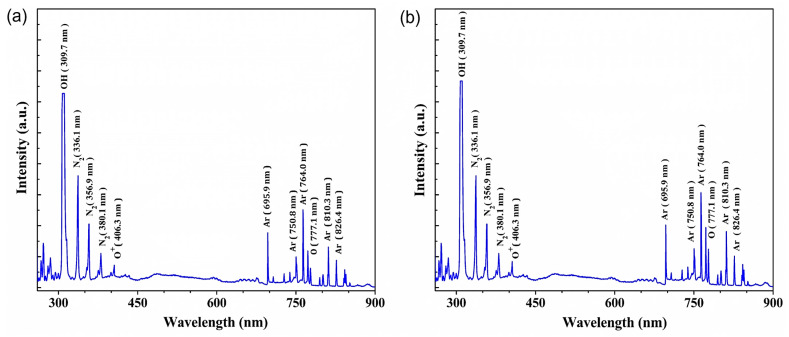
Representative OES of (**a**) argon gas and (**b**) Ar + O_2_ gas plasma.

**Figure 4 polymers-15-01711-f004:**
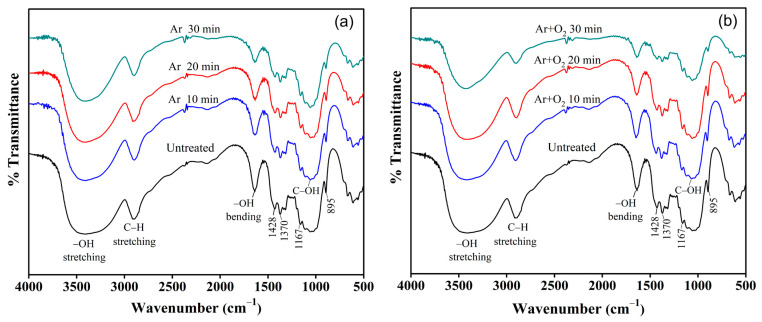
The FTIR spectra of bamboo fiber treated with (**a**) Ar gas and (**b**) Ar + O_2_ gas for various treatment times.

**Figure 5 polymers-15-01711-f005:**
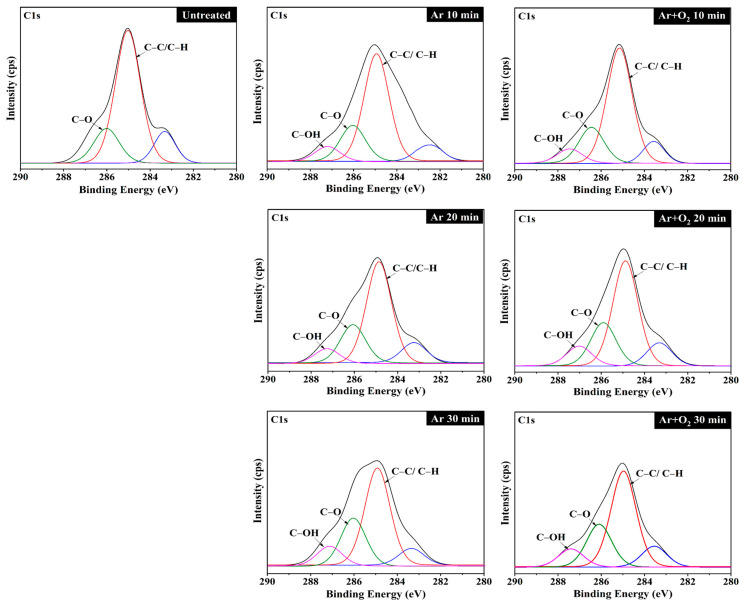
XPS spectra of BFs treated with Ar and Ar + O_2_ gas plasma for various treatment times.

**Figure 6 polymers-15-01711-f006:**
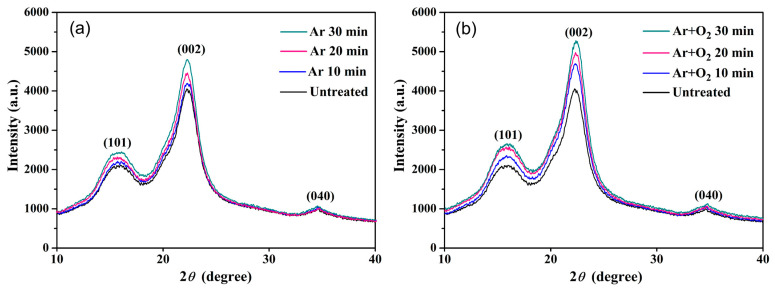
XRD diffractograms of untreated and treated BFs with (**a**) Ar gas and (**b**) Ar + O_2_ gas for various treatment times.

**Figure 7 polymers-15-01711-f007:**
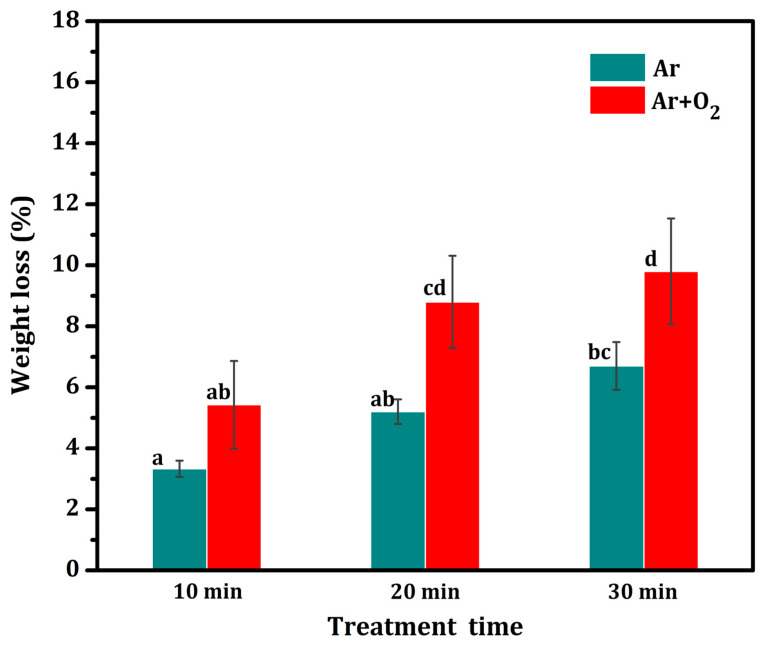
Comprehensively reduced weight loss percentage of fiber surface treated with plasma at various treatment times compared with the untreated fiber. Note: values indicated with the same letters are not significantly different at *p* ≤ 0.05.

**Figure 8 polymers-15-01711-f008:**
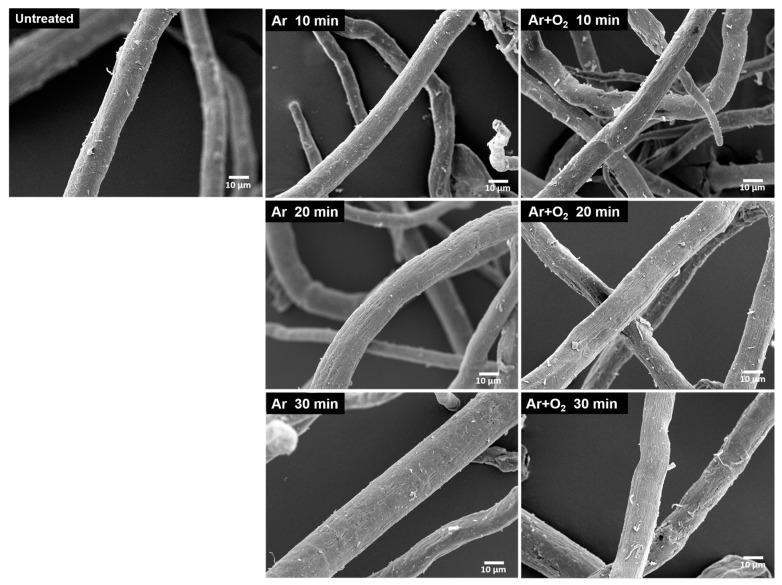
SEM images of the morphology of BFs treated with difference gas plasmas for various treatment times at 1000×.

**Figure 9 polymers-15-01711-f009:**
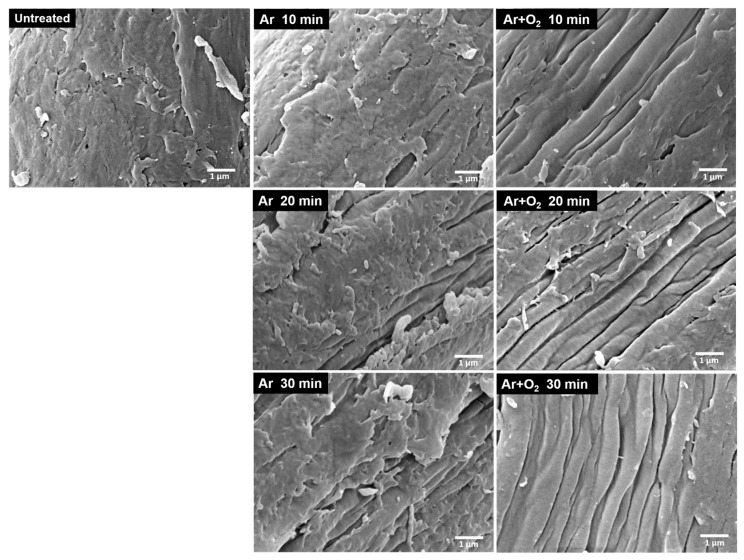
SEM images of the morphology of BFs treated with different gas plasmas for various treatment times at 15,000×.

**Figure 10 polymers-15-01711-f010:**
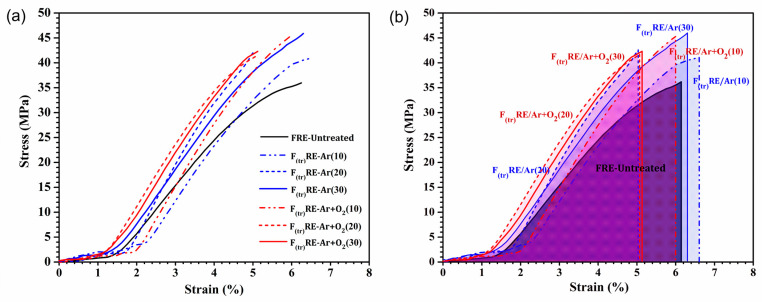
(**a**) Stress–strain curves and (**b**) calculated EA from the stress–strain curves of the FRE samples.

**Figure 11 polymers-15-01711-f011:**
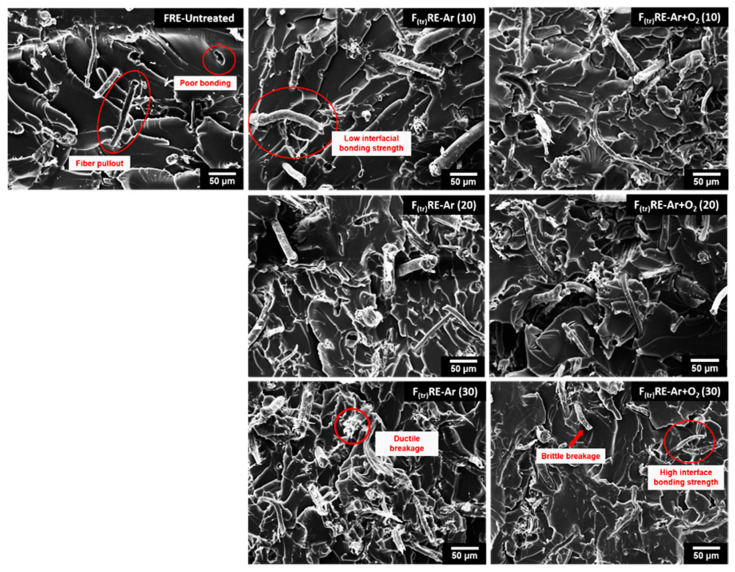
SEM images of fractured surfaces of BF/epoxy mixture samples with untreated fibers and fibers treated with Ar gas and Ar + O_2_ gas for various treatment times.

**Figure 12 polymers-15-01711-f012:**
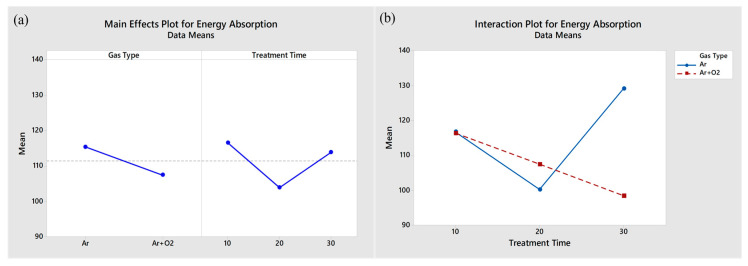
Main effect and interaction plot for the EA capacity for plasma-treated bamboo fiber/epoxy matrix composites.

**Table 1 polymers-15-01711-t001:** Plasma-treatment parameters for untreated and treated samples.

Conditions	Ar Gas Flow Rate (L/min)	O_2_ Gas Flow Rate (L/min)	Treatment Time (min)
Untreated	-	-	-
Ar 10 min	8	-	10
Ar 20 min	8	-	20
Ar 30 min	8	-	30
Ar + O_2_ 10 min	8	10	10
Ar + O_2_ 20 min	8	10	20
Ar + O_2_ 30 min	8	10	30

**Table 2 polymers-15-01711-t002:** The labels and composition of the FRE resin composite samples.

Treatments	Composition (wt/wt%)	
	Fiber	Epoxy Resin
FRE-Untreated	5	95
F_(tr)_RE-Ar (10)	5	95
F_(tr)_RE-Ar (20)	5	95
F_(tr)_RE-Ar (30)	5	95
F_(tr)_RE-Ar + O_2_ (10)	5	95
F_(tr)_RE-Ar + O_2_ (20)	5	95
F_(tr)_RE-Ar + O_2_ (30)	5	95

**Table 3 polymers-15-01711-t003:** Chemical constituents (%) of bamboo fiber.

Holocellulose (%)	Alpha-Cellulose (%)	Linin (%)	Extractive (%)	Ash (%)
73.10 ± 0.21	41.67 ± 0.35	28.88 ± 0.14	3.17 ± 0.07	2.04 ± 0.08

**Table 4 polymers-15-01711-t004:** Chemical composition of the untreated and treated BFs.

Treatments	Compositions (%) ^a,b^		O/C
	C	O	
Untreated	61.16	38.84	0.64
Ar 10 min	60.36	39.64	0.66
Ar 20 min	58.48	41.52	0.71
Ar 30 min	58.17	41.83	0.72
Ar + O_2_ 10 min	57.61	42.39	0.74
Ar + O_2_ 20 min	56.18	43.52	0.77
Ar + O_2_ 30 min	55.12	44.88	0.81

^a^ Atomic %. ^b^ C: carbon, O: oxygen.

**Table 5 polymers-15-01711-t005:** Surface area of bamboo fiber treated with plasma via BET experiment.

Type of Gas	Treatment Time (min)	BET Surface Area (m^2^/g)
Untreated	-	0.99 a
Ar	10 min	3.04 ab
Ar	20 min	3.96 abc
Ar	30 min	7.38 cd
Ar + O_2_	10 min	5.68 bcd
Ar + O_2_	20 min	8.86 d
Ar + O_2_	30 min	15.36 e

The mean values indicate that the different letters show significant difference according to LSD test at *p* ≤ 0.05.

**Table 6 polymers-15-01711-t006:** Mechanical properties of untreated FRE and plasma-treated F_(tr)_RE composites.

Sample	Tensile Strength (MPa) *	Elongation (%) *	Energy Absorption (MPa·mm/mm) *
FRE-untreated	36.94 ± 0.46 a	6.02 ± 0.49 ab	98.25 ± 11.78 a
F_(tr)_RE-Ar 10 min	39.80 ± 0.72 ab	6.77 ± 0.74 a	116.76 ± 15.65 ab
F_(tr)_RE-Ar 20 min	42.12 ± 1.11 bc	4.97 ± 0.45 b	100.30 ± 13.73 ab
F_(tr)_RE-Ar 30 min	46.30 ± 0.29 d	6.33 ± 0.21 ab	129.22 ± 5.28 b
F_(tr)_RE-Ar + O_2_ 10 min	45.19 ± 0.67 cd	5.92 ± 0.23 ab	116.38 ± 2.84 ab
F_(tr)_RE-Ar + O_2_ 20 min	40.95 ± 1.53 ab	5.09 ± 0.22 b	107.54 ± 7.94 ab
F_(tr)_RE-Ar + O_2_ 30 min	42.66 ± 0.59 bcd	5.12 ± 0.17 b	98.24 ± 4.43 a

* Values in the same column with the same letters are not significantly different at *p* ≤ 0.05.

## Data Availability

The data presented in this study are available on request from the corresponding author.

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
