# Peer review of "Surface Modification and Mechanical Properties Improvement of Bamboo Fibers Using Dielectric Barrier Discharge Plasma Treatment"

_polymers, 2023, doi:10.3390/polym15071711_

Round 1

Reviewer 1 Report

The authors presented an interesting experimental approach to the preparation and modification of BFs using DBD plasma treatment. The effect of argon gas (Ar) and oxygen gas (O2) as well as the treatment times on the properties of modified bamboo fibers was investigated. Characterization techniques such as SEM, XRD, XPS and FTIR were used to study, morphology, crystallinity, the chemical composition and characterize the surface chemical properties of the BFs. The specific surface areas were determined using (BET) method and the mechanical properties of fiber-reinforced epoxy (FRE) matrix composites were evaluated. The various experimental techniques used were sound and backed with numerous figures and supporting data.

However, there are a few minor comments which could make this manuscript in better shape.

1-     In the abstract:  the first sentence (line 25, 26) should be corrected.

2-     At the end of line 225: correct the word composition.

3-     The author prepared a fixed ratio of fiber and epoxy resin (5:95 w/wt%) for all the samples under study, so what is the purpose of table 2?

4-     Figure 6: to improve it; redraw the figure up to 2θ = 40o, increase the font size and Iam should be written in the specified position.

5-     The resolution of figure 12 needs to be improved.

Author Response

Journal: Polymer

Manuscript ID: polymers-2286450

Title: Surface Modification and Mechanical Properties Improvement of Bamboo Fibers Using Dielectric Barrier Discharge Plasma Treatment

RESPONSE TO REFEREES

We would like to thank the editors and reviewers for reading and giving us their encouraging comments. Overall, the authors revised the manuscript as kindly suggested by the editors and reviewers. In the manuscript, al the revisions were marked up using the “Track Changes” function.

Reviewer 1:

Comments and Suggestions for Authors

The authors presented an interesting experimental approach to the preparation and modification of BFs using DBD plasma treatment. The effect of argon gas (Ar) and oxygen gas (O2) as well as the treatment times on the properties of modified bamboo fibers was investigated. Characterization techniques such as SEM, XRD, XPS and FTIR were used to study, morphology, crystallinity, the chemical composition and characterize the surface chemical properties of the BFs. The specific surface areas were determined using (BET) method and the mechanical properties of fiber-reinforced epoxy (FRE) matrix composites were evaluated. The various experimental techniques used were sound and backed with numerous figures and supporting data.

However, there are a few minor comments which could make this manuscript in better shape.

  1. In the abstract: the first sentence (line 25, 26) should be corrected.

Ans: This sentence has been modified in line 25-27.                                                 

  1. At the end of line 225: correct the word composition.

Ans: The word composition has been revised in line 230.

  1. The author prepared a fixed ratio of fiber and epoxy resin (5:95 w/wt%) for all the samples under study, so what is the purpose of table 2?

Ans: Table 2 was used to indicate the label in each of the plasma-treated BFs conditions for the FRE samples and the title caption has been revised in line 238.

  1. Figure 6: to improve it; redraw the figure up to 2θ = 40o, increase the font size and Iam should be written in the specified position.

Ans: Figure 6 has been modified following the reviewer’s comment in line 372.

  1. The resolution of figure 12 needs to be improved.

Ans: The resolution of Figure 12 has been modified in line 496.

Reviewer 2 Report

The manuscript titled “Surface Modification and Mechanical Properties Improvement of Bamboo Fibers Using Dielectric Barrier Discharge Plasma Treatment” by Sawangrat, C.; et al. is an original work where the authors study the effect of plasma treatments with different ratio of argon and oxygen gases on the physico-chemical properties of bamboo fibers. Many complementary techniques were devoted to reach this purpose like fourier transform-infrared spectroscopy (FTIR), X-ray photoelectron spectroscopy (XPS), scanning electron microscopy (SEM), brunauer-emmett-teller (BET) and tensile tests. The authors found the best condition to treat bamboo fibers with plasma under argon for 30 minutes. The most relevant outcomes found in the present work could have a positive impact on many industrial sectors, such as agriculture, drug-delivery, and the design of more green-friendly materials, among others. The achieved results are well-discussed during the main body of the reported manuscript. The scientific paper is well written. In my opinion the present manuscript is innovative and the methodological approached used matches with the scope of Polymers. For the above described reasons, I will recommend the publication in Polymers once the following remarks are fixed:

--------

ABSTRACT

“The effect of argon gas (Ar) and oxygen (O2) (…)” (line 25). Please, the authors should modify this sentence by “The effect of argon (Ar) and oxygen (O2) gases (…)”.

“using dielectric barrier discharge (DBD) plasma generated power (…)” (lines 26-27). Please, this statement should be changed by “using dielectric barrier discharge (DBD) plasma at generated power (…)”.

(OPTIONAL) “X-ray photoelectron spectroscopy (…)” (line 32) and “Scanning electron microscopy (…)” (line 34). The authors could add the respective abbreviations “(XPS)” and “(SEM)” after each mentioned technique.

--------

INTRODUCTION

Introduction section is clear and concise. Some minor remarks must be addressed in order to increase the quality of the scientific content shown in the present manuscript:

(OPTIONAL) “depending on the materials’ composition (…)” (line 50). This sentence could be modified by “depending on the composition of the material (…)”.

“Four types of cold plasma discharge methods are commonly used for fiber-reinforced composites: corona discharge, atmospheric-pressure glow discharge (APGC), dielectric barrier discharge (DBD), and atmospheric-pressure plasma jet (APPK) [11]” (lines 59-62). Here, the authors present all existing plasma technologies reported in the use on fiber-reinforce composites and the citation 11 is relevant in this field. However, it would be desirable to add the following reference [1] to provide more insights of the use of cold plasma in the chemical modification of many others synthetic or natural fibers [1].

[1] Pillai, R.R.; et al. Plasma Surface Engineering of Natural and Sustainable Polymeric Derivatives and Their Potential Applications. Polymers 2023, 15, 400. https://doi.org/10.3390/polym15020400.

“Bamboo fibers (…) are mainly composed of cellulose, hemicellulose, lignin, and waxes [17]” (lines 70-72). Here, it may be desirable to specify the chemical composition of bamboo fibers as cellulose (36.8-54.9%), hemicellulose (62.0-79.9%), lignin (23.1-33.5%) and waxes (1.93-7.95%) [2]. This information will significantly aid to the potential readers the chemistry involved in the fibers used by the authors.

[2] Hartono, R.; et al. Physical, Chemical, and Mechanical Properties of Six Bamboo from Sumatera Island Indonesia and Its Potential Applications for Composite Materials. Polymers 2022, 14, 4868. https://doi.org/10.3390/polym14224868.

Then, after line 62, the authors should discuss alternative methodologies to oxidize plant fibers or lignocellulosic polymers like chemical 2,2,6,6-Tetramethylpeperidin-1-yl)oxyl (TEMPO) reaction [3] or by the Fenton reaction [4].

[3] Chitbanyong, K.; et al. Characterization of bamboo nanocellulose prepared by TEMPO-mediated oxidation. BioRes. 2018, 13, 4440-4454.

[4] Gerbin, E.; et al. Dual Antioxidant Properties and Organic Radical Stabilization in Cellulose Nanocomposite Films Functionalized by In Situ Polymerization of Coniferyl Alcohol. Biomacromolecules 2020, 21, 3163-3175. https://doi.org/10.1021/acs.biomac.0c00583.

“DBD plasma is a class of plasma sources (…)” (lines 96-97). Please, the authors should modify this sentence by “DBD is a class of plasma source (…)”.

“The morphology and microstructure (…), surface chemical properties (…), crystallinity (…), mechanical properties (…) model of the relationship (…)” (lines 109-117). Here, the authors should cite some relevant Review works where the detailed physico-chemical properties are addressed. In this context, [5], [6] and [7] should be cited for surface chemical properties, crystallinity and mechanical properties, respectively.

[5] Berthomieu, C.; et al. Fourier transform infrared (FTIR) spectroscopy. Photosynth. Res. 2009, 101, 157-170. https://doi.org/10.1007/s11120-009-9439-x.

[6] Ali, A.; et al. X-ray Diffraction Techniques for Mineral Characterization: A Review for Engineers of the Fundamentals, Applications and Research Directions. Minerals 2022, 12, 205. https://doi.org/10.3390/min12020205.

[7] Magazzù, A.; et al. Investigation of Soft Matter Nanomechanics by Atomic Force Microscopy and Optical Tweezers: A comprehensive Review. Nanomaterials 2023, 13, 963. https://doi.org/10.3390/nano13060963.

--------

MATERIALS AND METHODS

Materials and methods employed by the authors are unequivocally described which is crucial to mimic the same experimental approach in other labs placed in different locations. Only the following remarks should be fixed:

“The bamboo was peeled off (…)” (line 126). Please, the authors should change this sentence by “The bamboo fibers were peeled off (…)”.

Table 1 (line 181). Why did the authors not use the same flow rate of argon and oxygen gases when the tested conditions were both combined? Is there any particular reason?

Finally, the authors should specify the software used to analyze the raw data obtained by FTIR (section 2.6., lines 187-192), XPS (section 2.7., lines 193-199), XRD (section 2.8., lines 200-207), SEM (section 2.9., lines 208-214), BET (section 2.10., lines 215-220) and mechanical properties (section 2.11., lines 221-234). Please, the authors should change the abbreviation “(SM)” (line 208) by the broadly known “(SEM)”.

--------

RESULTS AND DISCUSSION

Authors perfectly state the most relevant outcomes found in the present work. Some points should be addressed to improve the manuscript quality.

I)        “Table 3 presents the chemical constituents of bamboo. The BFs consisted of 41.67% alpha-cellulose, 73.10% holocellulose, 28.88% lignin, 3.17% extractive, and 2.04 ash” (line 240-241). Please the authors should add at the end of this sentence the following statement “as previously reported [2]”.

II)     Figure 2 (line 258). The images of panels a-c should be rescaled at the same scan size (in this case with a lateral scale bar of 100 µm) and then, insert an inset in the right upper image with a zoomed area (scale bar of 10 µm). This fact could aid to better compare the observed morphology between the different tested conditions. Finally, Figures 8 (line 405) and 9 (line 408) show the magnification (1000x and 15000x, respectively). For this reason, the authors should also indicate the magnification in Figure 2 in order to be consistent.

III)  Figure 4 (line 308). Could the authors add the most significant chemical bonds associated for each wavenumber? (Similarly to the case of the hydroxyl groups).

IV)  “3,200-3,600 cm-1” (line 282). Authors should consider to erase the comma when the wavenumbers are referred. This comment should be extended for the rest of the section “3.4. Analysis of functional groups via FTIR” (lines 280-309).

V)     Maybe is a problem during the conversion process, in that case no actions are requested but otherwise, the authors should increase the lettering size of Figure 5 (line 347) because the displayed data cannot be clearly distinguished.

--------

CONCLUSIONS

The authors perfectly states this section. No actions are requested.

--------

REFERENCES

Bibliography citations are in the proper format of Polymers. The journal name should appear in abbreviated form. The authors should take care of this point.

--------

OVERVIEW AND FINAL COMMENTS

The submitted work is well-designed and the gathered results are interesting to better understand the positive impact of plasma oxidation of plant fibers in their surface interface adhesion and mechanical properties. This fact will aid to design more durable green-friendly composites. For these reasons, I will recommend the present scientific manuscript for further publication in Polymers once all the aforementioned suggestions will be properly fixed.

Author Response

Journal: Polymer

Manuscript ID: polymers-2286450

Title: Surface Modification and Mechanical Properties Improvement of Bamboo Fibers Using Dielectric Barrier Discharge Plasma Treatment

RESPONSE TO REFEREES

We would like to thank the editors and reviewers for reading and giving us their encouraging comments. Overall, the authors revised the manuscript as kindly suggested by the editors and reviewers. In the manuscript, al the revisions were marked up using the “Track Changes” function.

Reviewer 2

Comments and Suggestions for Authors

The manuscript titled “Surface Modification and Mechanical Properties Improvement of Bamboo Fibers Using Dielectric Barrier Discharge Plasma Treatment” by Sawangrat, C.; et al. is an original work where the authors study the effect of plasma treatments with different ratio of argon and oxygen gases on the physico-chemical properties of bamboo fibers. Many complementary techniques were devoted to reach this purpose like fourier transform-infrared spectroscopy (FTIR), X-ray photoelectron spectroscopy (XPS), scanning electron microscopy (SEM), brunauer-emmett-teller (BET) and tensile tests. The authors found the best condition to treat bamboo fibers with plasma under argon for 30 minutes. The most relevant outcomes found in the present work could have a positive impact on many industrial sectors, such as agriculture, drug-delivery, and the design of more green-friendly materials, among others. The achieved results are well-discussed during the main body of the reported manuscript. The scientific paper is well written. In my opinion the present manuscript is innovative and the methodological approached used matches with the scope of Polymers. For the above describe reasons, I will recommend the publication in Polymers once the following remarks are fixed:

ABSTRACT

  1. “The effect of argon gas (Ar) and oxygen (O2) (…)” (line 25). Please, the authors should modify this sentence by “The effect of argon (Ar) and oxygen (O2) gases (…)”.

Ans: This sentence has been revised in line 25.

  1. “using dielectric barrier discharge (DBD) plasma generated power (…)” (lines 26-27). Please, this statement should be changed by “using dielectric barrier discharge (DBD) plasma at generated power (…)”.

Ans: This sentence has been revised in line 26-27.

  1. (OPTIONAL) “X-ray photoelectron spectroscopy (…)” (line 32) and “Scanning electron microscopy (…)” (line 34). The authors could add the respective abbreviations “(XPS)” and “(SEM)” after each mentioned technique.

Ans: This sentence has been modified following the reviewer’s comment in line 29-35.

INTRODUCTION

Introduction section is clear and concise. Some minor remarks must be addressed in order to increase the quality of the scientific content shown in the present manuscript:

  1. (OPTIONAL) “depending on the materials’ composition (…)” (line 50). This sentence could be modified by “depending on the composition of the material (…)”.

Ans: This sentence has been modified following the reviewer’s comment in line 50.

  1. “Four types of cold plasma discharge methods are commonly used for fiber-reinforced composites: corona discharge, atmospheric-pressure glow discharge (APGC), dielectric barrier discharge (DBD), and atmospheric-pressure plasma jet (APPK) [11]” (lines 59-62). Here, the authors present all existing plasma technologies reported in the use on fiber-reinforce composites and the citation 11 is relevant in this field. However, it would be desirable to add the following reference [1] to provide more insights of the use of cold plasma in the chemical modification of many others synthetic or natural fibers [1].

[1] Pillai, R.R.; et al. Plasma Surface Engineering of Natural and Sustainable Polymeric Derivatives and Their Potential Applications. Polymers 2023, 15, 400. https://doi.org/10.3390/polym15020400.

       Ans: The reference has been added in the citation 12 (line 62).

  1. “Bamboo fibers (…) are mainly composed of cellulose, hemicellulose, lignin, and waxes [17]” (lines 70-72). Here, it may be desirable to specify the chemical composition of bamboo fibers as cellulose (36.8-54.9%), hemicellulose (62.0-79.9%), lignin (23.1-33.5%) and waxes (1.93-7.95%) [2]. This information will significantly aid to the potential readers the chemistry involved in the fibers used by the authors.

[2] Hartono, R.; et al. Physical, Chemical, and Mechanical Properties of Six Bamboo from Sumatera Island Indonesia and Its Potential Applications for Composite Materials. Polymers 2022, 14, 4868. https://doi.org/10.3390/polym14224868.

       Ans: The reference has been added in the citation 19 (line 70-72).

  1. Then, after line 62, the authors should discuss alternative methodologies to oxidize plant fibers or lignocellulosic polymers like chemical 2,2,6,6-Tetramethylpeperidin-1-yl)oxyl (TEMPO) reaction [3] or by the Fenton reaction [4].

[3] Chitbanyong, K.; et al. Characterization of bamboo nanocellulose prepared by TEMPO-mediated oxidation. BioRes. 2018, 13, 4440-4454.

[4] Gerbin, E.; et al. Dual Antioxidant Properties and Organic Radical Stabilization in Cellulose Nanocomposite Films Functionalized by In Situ Polymerization of Coniferyl Alcohol. Biomacromolecules 2020, 21, 3163-3175. https://doi.org/10.1021/acs.biomac.0c00583.

Ans: The information has been modified according to the reviewer's advice in line 72-77 and the reference has been added in the citation 20 and 21.

  1. “DBD plasma is a class of plasma sources (…)” (lines 96-97). Please, the authors should modify this sentence by “DBD is a class of plasma source (…)”.

Ans: This sentence has been revised in line 101.

  1. “The morphology and microstructure (…), surface chemical properties (…), crystallinity (…), mechanical properties (…) model of the relationship (…)” (lines 109-117). Here, the authors should cite some relevant Review works where the detailed physico-chemical properties are addressed. In this context, [5], [6] and [7] should be cited for surface chemical properties, crystallinity and mechanical properties, respectively.

[5] Berthomieu, C.; et al. Fourier transform infrared (FTIR) spectroscopy. Photosynth. Res. 2009, 101, 157-170. https://doi.org/10.1007/s11120-009-9439-x.

[6] Ali, A.; et al. X-ray Diffraction Techniques for Mineral Characterization: A Review for Engineers of the Fundamentals, Applications and Research Directions. Minerals 2022, 12, 205. https://doi.org/10.3390/min12020205.

[7] Magazzù, A.; et al. Investigation of Soft Matter Nanomechanics by Atomic Force Microscopy and Optical Tweezers: A comprehensive Review. Nanomaterials 2023, 13, 963. https://doi.org/10.3390/nano13060963.

     Ans: The references have been added in the citation 40-42, respectively (line 116-120)   

MATERIALS AND METHODS

Materials and methods employed by the authors are unequivocally described which is crucial to mimic the same experimental approach in other labs placed in different locations. Only the following remarks should be fixed:

  1. “The bamboo was peeled off (…)” (line 126). Please, the authors should change this sentence by “The bamboo fibers were peeled off (…)”.

Ans: This sentence has been revised following the reviewer’s comment in line 131.

  1. Table 1 (line 181). Why did the authors not use the same flow rate of argon and oxygen gases when the tested conditions were both combined? Is there any particular reason?

Ans: In this experiment, argon gas was aimed to ignite the plasma probe of DBD only in which from our preliminary test, only the flow rate of 8 L/min is enough for plasma ignition. Moreover, in preliminary test, we also noticed that if we used the same flow rate of argon and oxygen gases, it resulted in nonuniformity of plasma ignition.

  1. Finally, the authors should specify the software used to analyze the raw data obtained by FTIR (section 2.6., lines 187-192), XPS (section 2.7., lines 193-199), XRD (section 2.8., lines 200-207), SEM (section 2.9., lines 208-214), BET (section 2.10., lines 215-220) and mechanical properties (section 2.11., lines 221-234). Please, the authors should change the abbreviation “(SM)” (line 208) by the broadly known “(SEM)”.

Ans:  The software information of each characterize method has been added base on the reviewer’s advice, for the FTIR in line 193-194, XPS in line 202 , XRD in line 206 , BET in line 223-224 , and a universal testing machine in line 235.

RESULTS AND DISCUSSION

Authors perfectly state the most relevant outcomes found in the present work. Some points should be addressed to improve the manuscript quality.

  1. “Table 3 presents the chemical constituents of bamboo. The BFs consisted of 41.67% alpha-cellulose, 73.10% holocellulose, 28.88% lignin, 3.17% extractive, and 2.04 ash” (line 240-241). Please the authors should add at the end of this sentence the following statement “as previously reported [2]”.

Ans: This sentence has been revised in line 246-247 and the reference has been added in the citation 19 at the end of sentence.

  1. Figure 2 (line 258). The images of panels a-c should be rescaled at the same scan size (in this case with a lateral scale bar of 100 µm) and then, insert an inset in the right upper image with a zoomed area (scale bar of 10 µm). This fact could aid to better compare the observed morphology between the different tested conditions. Finally, Figures 8 (line 405) and 9 (line 408) show the magnification (1000x and 15000x, respectively). For this reason, the authors should also indicate the magnification in Figure 2 in order to be consistent.      

Ans: Figure 2 has been rescaled at the same scan size by calculating based on the magnification used, which has been added to the figure caption in line 263.

  1. Figure 4 (line 308). Could the authors add the most significant chemical bonds associated for each wavenumber? (Similarly to the case of the hydroxyl groups).

Ans: The most significant chemical bonds have been added into Figure 4, as shown in line 307.

  1. “3,200-3,600 cm-1” (line 282). Authors should consider to erase the comma when the wavenumbers are referred. This comment should be extended for the rest of the section “3.4. Analysis of functional groups via FTIR” (lines 280-309).

Ans: The FTIR analysis result has been revised in line 288-293, and 300.

  1. Maybe is a problem during the conversion process, in that case no actions are requested but otherwise, the authors should increase the lettering size of Figure 5 (line 347) because the displayed data cannot be clearly distinguished.

Ans: The lettering size in Figure 5 has been modified to be clearly distinguished, as shown in line 348.

CONCLUSIONS

The authors perfectly state this section. No actions are requested.

REFERENCES

Bibliography citations are in the proper format of Polymers. The journal name should appear in abbreviated form. The authors should take care of this point.

Ans: The citation section has been modified in an abbreviated form following the reviewer's comment.

OVERVIEW AND FINAL COMMENTS

The submitted work is well-designed and the gathered results are interesting to better understand the positive impact of plasma oxidation of plant fibers in their surface interface adhesion and mechanical properties. This fact will aid to design more durable green-friendly composites. For these reasons, I will recommend the present scientific manuscript for further publication in Polymers once all the aforementioned suggestions will be properly fixed.

Round 2

Reviewer 2 Report

The authors have done a great volume of effort in order to improve the manuscript scientific quality and for this reason, I consider this work appropiate for further publication in Polymers journal.